# Two-Stage Oxidative Leaching of Low-Grade Copper–Zinc Sulfide Concentrate

**DOI:** 10.3390/microorganisms10091781

**Published:** 2022-09-03

**Authors:** Aleksandr Bulaev, Vitaliy Melamud

**Affiliations:** Research Center of Biotechnology, Russian Academy of Sciences, 119071 Moscow, Russia

**Keywords:** bioleaching, ferric leaching, chloride leaching, chalcopyrite, tennantite, sphalerite, two-stage processes

## Abstract

Bioleaching may be effectively used to extract nonferrous metals from sulfide ores and concentrates. At the same time, some minerals are refractory and their bioleaching rate is often comparatively low that does not allow the required metal extraction rate to be achieved. In the present work, we studied the two-stage process, which included stages of biological and chemical leaching, to improve copper extraction from low grade Cu–Zn sulfide concentrate containing chalcopyrite, tennantite, pyrite, and sphalerite. Bioleaching was conducted in the continuous mode in three laboratory scale reactors connected in series. The pulp density was 10% and the residence time was 7 days. The temperature was 40 °C in the 1st reactor and 50 °C in the 2nd and 3rd reactors. Bioleaching allowed the extraction of 29.5 and 78% of Cu and Zn, respectively. The solid bioleach residue obtained was then treated for additional Cu and Zn recovery using high temperature leaching at 90 °C for 25 h. The liquid phase of the bioleaching pulp contained Fe^3+^ ions, which is the strong oxidant, and the leach solution was supplemented with NaCl. In the presence of the maximal NaCl concentration (1 M), Cu and Zn extraction reached 48 and 84%. Thus, two-stage leaching may allow to increase bioleaching efficiency and may be used to improve the bioleaching rate of refractory minerals, such as chalcopyrite.

## 1. Introduction

Most of the copper in the world is produced by processing sulfide ores by flotation to obtain copper concentrates and their subsequent processing using pyrometallurgical technologies [1,2,3,4]. The most abundant copper mineral that plays a crucial role in its production is chalcopyrite (CuFeS_2_). Pyrometallurgical processing of sulfide concentrates results in the formation of gas emissions containing SO_2_, which is a certain problem for the environment [2,5,6]. Sulfur dioxide can be used to produce sulfuric acid, which is a by-product of copper production, and the gas produced after sulfur dioxide removal can be safely released into the environment [5]. A greater problem in the pyrometallurgical processing of sulfide concentrates is the presence of various impurities. For example, copper concentrates may contain elements such as arsenic, antimony, lead, mercury, and bismuth. During the pyrometallurgical processing, these components can pose a problem for the production of copper, both due to an increase in negative environmental effects and due to the impact on the quality of metallurgical products. Arsenic, antimony, and mercury pass into the gas phase during the processing, which leads to the need for additional purification of gas emissions from these components [7,8,9]. In addition, the presence of these elements in raw materials and products of its flotation reduces the quality of the resulting copper [9,10,11,12]. Other impurities can also significantly complicate the processing of copper concentrates and reduce its efficiency. A high zinc content in copper concentrates, obtained from polymetallic ores, reduces copper extraction during pyrometallurgical processes, increasing slag viscosity [8,12]. In addition, one of the main problems of copper production is the formation of flotation tailings and slags during ore dressing and smelting/converting. Tailing and slug storage possesses a harmful environmental effect due to the high content of iron, non-ferrous metals, and toxic metalloids, as well as toxic drainage formation, that leads to element migration [13,14].

The application of various hydrometallurgical technologies is an alternative for pyrometallurgical technologies. In the case of copper production, pyrometallurgical technologies usually provide greater efficiency in the processing of raw materials and greater productivity, however, hydrometallurgical technologies can ensure the processing of low-grade ores, as well as substandard concentrates, including those containing harmful impurities [4,9,15,16,17,18]. Different hydrometallurgical technologies exist that make it possible to extract copper into a liquid phase, and then obtain commercial metal using solvent extraction and electrolysis [18]. Among the technologies that have been commercialized, there are acid leaching, autoclave leaching, bioleaching, as well as atmospheric leaching using various oxidizing agents (oxygen, Fe^3+^ ions), which can be used for raw materials with different mineral compositions (oxidized and sulfide ores, as well as concentrates) [18]. For example, acid leaching has been successfully used to process oxidized and mixed ores, pressure oxidation is usually used to process sulfide concentrates, and bioleaching is widely used to process low-grade sulfide ores (heap bioleaching) and finds limited use for processing substandard copper concentrates [18,19,20,21,22,23,24,25]. It should be noted that the presence of toxic impurities, such as arsenic, in ores and concentrates can provide a competitive advantage for the use of hydrometallurgical technologies, since they allow the performance of the extraction of metal without the formation of gas emissions. Therefore, hydrometallurgical processing of arsenic-containing ores and concentrates is an urgent topic for research and finds practical application [8,9].

The methods based on oxidative leaching can be successfully applied to extract copper and other non-ferrous metals from various sulfide ores and concentrates. Heap bioleaching has been successfully used to process low-grade sulfide ores of non-ferrous metals [18,19] and can also be used for bioleaching of copper and other metals from ores containing arsenic, including sulfide and arsenide ores [11,26,27]. Stirred tank reactor bioleaching is more often used to process refractory gold concentrates, which usually contain arsenopyrite (FeAsS), which makes it difficult to process them using pyrometallurgical technologies (roasting) [23]. In addition, it was shown that reactor bioleaching can be successfully used for processing sulfide concentrates containing Co, Ni, Cu, and Zn [19,28,29]. In particular, reactor bioleaching can be successfully used (at least on a pilot scale) for the processing of low-grade copper–zinc concentrate, in which it was irrational to use pyrometallurgical processing due to the high zinc content [19]. There are currently no examples of using reactor bioleaching to process arsenic-containing copper concentrates on an industrial scale, but there is an enterprise that uses reactor bioleaching to recover nickel and cobalt from a sulfide concentrate containing about 1.5% arsenic. It cannot be treated using pyrometallurgical methods due to the high content of toxic components, while reactor bioleaching makes it possible to extract non-ferrous metals without the formation of gas emissions [29].

It is worth considering that some copper minerals, including arsenic-containing ones, may be resistant to bioleaching compared to other minerals [30,31,32]. It was shown that, in terms of resistance to bioleaching, the minerals contained in the concentrate could be ranked in the following order:

Chalcocite > bornite > cubanite > covellite > enargite > chalcopyrite.

Thus, compared to a number of other common copper sulfide minerals, enargite has been shown to be more resistant to bioleaching. Moreover, our previous studies on the reactor bioleaching of copper–zinc concentrates, which contained pyrite (FeS_2_), chalcopyrite (CuFeS_2_), tennantite (Cu_12_As_4_S_13_), and sphalerite (ZnS) at 40 °C [33,34], demonstrated that tennantite was more resistant to bioleaching than chalcopyrite, and the content of arsenic and tennantite in solid residues of bioleaching increased compared to the oxidizable concentrate.

Atmospheric leaching (i.e., leaching with various oxidizing agents at atmospheric pressure) is also used to process copper ores and concentrates. A number of research works and examples of practical applications have shown that leaching with sulfuric acid solutions of ferric sulfate can be promising for copper extraction [17,18,35,36]. This process involves the leaching of copper from sulfide minerals using Fe^3+^ ions, which is a strong oxidizing agent, in stirred reactors at atmospheric pressure and elevated temperatures (80–90 °C). At the same time, the process of generation of Fe^3+^ ions, which are reduced during the oxidation of sulfide minerals, is important in this technology. With the technology used at the Sepon factory (Laos), atmospheric leaching with Fe^3+^ ions at 80 °C allows to extract copper from a concentrate containing pyrite, covellite, enargite, and chalcopyrite [17,18]. Atmospheric leaching makes it possible to extract the main part of copper from the concentrate, while the solid residue, which contains pyrite and sulfur as well as residual amounts of copper, is processed by autoclave leaching, allowing generation of sulfuric acid and ferric sulfate, that are then used as a leaching agent, as well as recovery of the remaining copper. At the Las Cruces deposit (Spain), ferric sulfate leaching is used to extract copper from ore, which contains an average of 6.2% copper in the form of digenite, chalcocite, covellite, and chalcopyrite [18,34,35,36,37]. The leaching is carried out at 90 °C, which allows the recovery of about 92% of the copper for 8 h. During the development of technologies, various options for the generation of an oxidant were assumed, such as autoclave oxidation or biooxidation of ferrous ions Fe^2+^ [35]. However, when the technology was introduced on an industrial scale, it was found that the best option is the chemical oxidation of Fe^2+^ ions by supplying oxygen directly to the reactors in which the concentrate is leached [38]. Thus, there are examples of successful application of chemical leaching using ferric sulfate as an oxidizing agent; however, it should be noted that the implementation of such processes requires regeneration of the oxidizing agent (Fe^3+^ ions). Since the oxidation of sulfide minerals requires a large amount of ferric sulfate, ferric ions regeneration complicates the technological schemes and increases the cost of processing sulfide concentrates.

There are data on the study of enargite and tennantite leaching processes using Fe^3+^ [38]. It was shown that the presence of Fe^3+^ ions increased the rate of enargite leaching in sulfuric acid solutions, while in their absence in sulfuric acid solutions, the enargite leaching rate was very low [38,39]. In ref. [40], leaching of tennantite, tetrahedrite, and enargite in solutions of sulfate and ferric chloride was studied. In this work, it was also shown that the leaching rate of the studied minerals also depended on temperature and the leaching rate was low at temperatures below 100 °C. In ref. [41], the process of enargite leaching in the presence of pyrite was studied. It was shown that the rate of enargite leaching in iron (III) sulfate solution depended on the presence of pyrite and increased with an increase in the ratio of pyrite and enargite in the pulp. The authors of [42] studied the process of bioleaching of enargite and tennantite in the presence of ferric sulfate, which was additionally introduced into the medium. It was shown that when using a medium containing additional iron ions, bioleaching was faster than using a medium in which ferrous sulfate was not added.

Thus, the analysis of literature data shows that for the leaching of copper from ores and concentrates containing minerals such as chalcopyrite, enargite, and tennantite, methods based on various processes of oxidative leaching of sulfide minerals (bioleaching, leaching in solutions of ferric ions) may be used. At the same time, it was shown that enargite and tennantite, as well as chalcopyrite, can be more resistant in leaching processes, and the limited experience in the practical application of these technologies does not allow us to fully assess the potential for their widespread implementation. The main problems that may limit the practical use of technologies based on oxidative leaching for the processing of copper ores and concentrates containing refractory copper sulfide minerals are the relatively low rates of leaching of these minerals under normal conditions, as well as the need to use equipment that allows working at high temperatures or pressures. A high content of other elements in copper concentrates, such as zinc, is also a problem, as well as the impossibility of obtaining concentrates with a high copper content from ores [8,19]. Due to the fine dissemination of minerals in polymetallic ores, for example, copper–zinc, as well as the fine intergrowth of copper and zinc sulfides and pyrite, the problem of treatment of polymetallic ores, from which it is impossible to obtain conditioned non-ferrous metal concentrates, is of great importance. At the same time, reactor bioleaching has been successfully used for the processing of polymetallic concentrates containing Cu and Zn [19]. One of the challenges for applying bioleaching to these concentrates is the fact that many copper sulfide minerals, such as chalcopyrite and tennantite, are relatively resistant to bioleaching [33,34,43]. Thus, it may be promising to develop combined technological schemes that will allow efficient processing of sulfide concentrates of complex mineral composition at a higher rate than already known processes.

In addition to oxidative leaching processes based on copper and other metal leaching in ferric systems, various approaches for copper leaching have also been developed, including those based on application ammoniacal lixiviants, ionic liquids, glycine, hydrogen peroxide, MnO_2_, and nitrate [44,45,46]. Despite this, processes based on application on leaching in ferric systems (including leaching with ferric sulfate and chloride, bioleaching) are still considered as a promising method for sulfide concentrate treatment due to their comparative cheapness and simplicity, well-known processes of further copper extraction from pregnant leaching solutions, and the possibility to extract other valuable metals simultaneously with copper (zinc, cobalt, and nickel).

The purpose of this work was to conduct laboratory tests on tank bioleaching of a copper–zinc concentrate containing chalcopyrite, tennantite, sphalerite, and pyrite, as well as to develop a two-stage hydrometallurgical scheme, including (1) the bioleaching stage for copper and zinc extraction and to obtain a solution containing the leaching agent (ferric ions) and (2) the high-temperature chemical oxidation stage to increase the metal extraction degree. According to the proposed scheme, these two stages may be used as sequential stages of the integrated continuous leaching process.

## 2. Materials and Methods

### 2.1. Composition of the Flotation Concentrate Sample

The sample of copper–zinc sulfide concentrate was provided by Ural Mining and Metallurgical Company (Verknaya Pyshma, Russia). The main elemental contents in the sample were determined using the phase analysis method in the testing laboratory “Gintsvetmet-Analytics” (Moscow, Russia) [47] and are presented in Table 1.

The mineral composition of the concentrate as well as leaching residues was determined using an XRD-6000 diffractometer (Shimadzu, Kyoto, Japan). The main sulfide minerals in the sample were chalcopyrite (CuFeS_2_), tennantite (Cu_12_As_4_S_13_), sphalerite (ZnS), and pyrite (FeS_2_) (Figure 1).

### 2.2. Microbial Population and Its Analysis

To perform bioleaching experiments, a mixed culture of acidophilic microorganisms capable of iron and sulfur oxidation was used including *Acidithiobacillus caldus*, *Sulfobacillus benefaciens*, *Leptospirillum ferriphilum*, *Ferroplasma acidiphilum*, *Cuniculiplasma divulgatum*, and *Acidiplasma* sp. This culture was used in our previous work to perform stirred tank bioleaching of the same concentrate at 40 °C [33].

As the composition of microbial populations is usually changed during long-term continuous bioleaching tests performed under different conditions, in the present work, we analyzed compositions of the populations formed in the experiments with different temperatures and residence time. For this purpose, we used metabarcoding of the hypervariable V3–V4 region of the 16S rRNA gene carried out using the MiSeq system (Illumina, San Diego, CA, USA). The biomass samples were collected at the end of each experiment. The biomass collection using centrifugation, DNA isolation, sequencing, and data analysis procedures were described in detail in our previous articles [33,34].

### 2.3. Experimental Setup and Biooxidation

Continuous bioleaching tests were carried out in laboratory scale stirred tank reactors with a working volume of 1.5 L. The conditions were the following: stirring rate was 500 rpm, aeration rate was 5 L/min, and pulp density was 10%. Bioleaching tests were carried out in three reactors connected in series. In the 1st reactor, residence time was 3 days, while in the 2nd and in the 3rd reactors, it was 2 days. Thus, the total residence time was 7 days.

In the first temperature mode, the temperature in all reactors was 40 °C. In the second temperature mode, the temperature in the 1st reactor was 40 °C, while in the 2nd and 3rd reactors it was 50 °C.

To maintain the temperature required and perform the stirring, TW-2.03 circulating water baths (Elmi, Riga, Latvia) and U-shaped titanium heat exchangers, as well as RW20 overhead stirrers (IKA, Staufen, Germany), were used.

The mineral nutrient medium with the following composition (g/L) was used in the bioleaching tests: (NH_4_)_2_SO_4_—0.750; KCl—0.050; MgSO_4_ × 7H_2_O—0.125; K_2_HPO_4_—0.125; distilled water—1.0 L. In our previous works, it was shown that this medium may be used for bioleaching processes [33].

In the present study, bioleaching was started under conditions similar to those used in our previous trials [33]; therefore, when the experiments of [33] were completed, the pulp obtained was used as an inoculum to perform experiments in the present work.

### 2.4. Chemical Leaching

After biooxidation, the solid residue of the concentrate bioleaching process was subjected to high temperature leaching to achieve a more complete oxidation of the sulfide minerals. The process of chemical oxidation was carried out in a laboratory scale titanium stirred tank reactor with a working volume of 2.5 L. To maintain the temperature required and perform the stirring, an ED-5 heating circulator (Julabo, Seelbach, Germany) with U-shaped titanium heat exchangers and an RW 20 overhead stirrer (IKA, Staufen, Germany) were used. The leaching was performed at 90 °C and with a stirring rate of 500 rpm in a batch mode.

The duration of the chemical oxidation process was 25 h. To carry out the chemical leaching process, 1 L of pulp obtained after biooxidation in the second temperature mode was placed in a titanium reactor and heated to 90 °C. To determine the effect of sodium chloride ions on the leaching, 0.5 and 1 M (29 and 58 g/L, respectively) sodium chloride was added to the pulp before high-temperature leaching, since it was previously shown that the presence of chloride ions in the medium makes it possible to increase the efficiency of chalcopyrite oxidation [45,48,49].

Thus, as the liquid phase of the bioleaching pulp contained Fe^3+^ ions, which are the strong oxidants, and, in some cases, were supplemented with NaCl, ferric and chloride ions were leaching agents in the chemical leaching process.

### 2.5. Sampling and Chemical Analysis

To assess process activity and determine metal extraction, the key parameters of the liquid phase were determined.

The pH and redox potential (Eh) were measured in the liquid phase using a pH-150MI pH meter (Izmeritelnaya tekhnika, Moscow, Russia) with thermocompensation. Fe^3+^ and Fe^2+^ ions were determined using trilonometric titration [50]. Cu and Zn concentrations in the solutions were determined using a Perkin Elmer 3100 flame atomic absorption spectrometer (Perkin Elmer, Waltham, MA, USA). Extractions of Cu and Zn were calculated using the concentration of Cu and Zn ions in pregnant solutions.

### 2.6. Data Processing

The results obtained were processed using MS 15.0.459.1506 Excel 2013 software (Microsoft, Redmond, WA, USA). Average values of the parameters are shown.

## 3. Results

### 3.1. Bioleaching 

The parameters of the liquid phase of the reactor pulp during biooxidation under different conditions are presented in Table 2.

The pH level of the pulp in all reactors was close to or below 1, and no addition of sulfuric acid was required to maintain the pH of the pulp. The redox potential was high (above 745 mV) and increased from the first to the third reactor in both experiments, while ferrous iron in the liquid phase was detected in trace amounts. This indicates high activity of microorganisms performing bioleaching. Despite this, the leaching rate of the metals differed (Table 3).

In the first temperature mode, zinc extraction was at a maximum in the second reactors reaching about 90%, while in the second temperature mode it was lower and reached a maximum already in the first reactor and further bioleaching did not increase zinc extraction.

Copper extraction was lower compared to zinc in all reactors that is explained by the fact that sphalerite is less refractory to bioleaching compared to copper minerals, chalcopyrite, and tennantite [34,43]. In contrast to zinc, copper extraction in the second temperature mode was insignificantly higher, probably due to the temperature effect [16,17], and increased from the first to the third reactor in both temperature modes.

Although arsenic was partially leached during the bioleaching, its content in biooxidation residues increased in comparison to the concentrate and reached 4.0 and 3.6% in the residue obtained in the first and in the second temperature modes, respectively. The tennantite content in bioleaching residues increased in comparison to the concentrate in contrast to chalcopyrite (Figure 1), that may be explained by its higher resistance to biooxidation and lower dependence of its leaching rate on temperature compared to chalcopyrite [33,42].

Thus, a temperature increase led to the increase in copper extraction, probably due to the chalcopyrite leaching rate, while zinc extraction at a higher temperature decreased. Based on the results obtained, the reliable explanation of the zinc recovery decrease during the bioleaching at a higher temperature may not be obtained. In our previous work [34], we observed the same effect during the bioleaching of similar concentrates at different temperatures. This effect may be explained by similar trends in the changes in microbial population composition (the decrease in *Sulfobacillus* and *Acidithiobacillus* fraction and the increase in the proportion of archaea) when increasing the temperature from 40 to 50 °C (Figure 2 and Section 3.2). However, as information on the effect of the change in microbial population composition on sphalerite leaching is absent, this explanation may be used only as assumption.

### 3.2. Microbial Population Analysis

The analysis of the microbial population composition made it possible to evaluate the influence of the difference in temperature on acidophilic microorganisms (Figure 2).

The populations in the first reactors differed in the first and in the second temperature modes despite the same process conditions and similar parameters of the pulp obtained (Table 2). Representatives of the genera *Acidithiobacillus*, *Sulfobacillus*, and *Ferroplasma* were predominant, while the proportions of these genera differed. Additionally, the abundance of the archaea *Cuniculiplasma* in the second experiment was about 10 times higher. Thus, small fluctuations in the physicochemical parameters of the pulp, which are inevitable in the steady state of the process, could lead to a significant change in the composition of the microbial community.

In the first temperature mode, the abundance of the bacteria *Sulfobacillus* as well as archaea of the genera *Ferroplasma* and *Cuniculiplasma* increased from the first to the third reactor, while the abundance of the genera *Acidithiobacillus* decreased.

A similar increase in the abundance of the archaea of the order *Thermoplasmatales* from the first to the third reactor was observed in the second mode. Similarly, other groups of the order *Acidiplasma* as well as uncultivated archaea A-plasma were predominant instead of *Ferroplasma* at 50 °C, while the abundance of *Cuniculiplasma* was high in both temperature modes.

The predominance of the bacteria of the genus *Sulfobacillus* and the archaea of the genus *Ferroplasma* in the microbial population of the third reactor at 40 °C can be explained by the accumulation of the metabolites of autotrophic microorganisms (*Acidithiobacillus*) in the pulp that led to the predominance of bacteria of the genus *Sulfobacillus* and archaea of the genus *Ferroplasma*, which are generally able to use organic matter (including metabolites) as a source of carbon and energy [51,52,53]. Similarly, the predominance of the genera *Acidiplasma* and *Cuniculiplasma* in the second temperature mode may be explained, as representatives of both genera require organic compounds for the growth [51,52,54,55,56].

### 3.3. Chemical Leaching

The results of chemical leaching are summarized in Figure 3 and Figure 4 and Table 4. The presented results show that chemical leaching at 90 °C without sodium chloride did not increase the level of metal extraction, and zinc concentration in this case decreased during the leaching. The addition of 0.5 M NaCl insignificantly increased copper extraction, while zinc concentration also decreased. The addition of 1 M NaCl to the pulp during chemical leaching led to a significant increase in the degree of extraction of both metals into solution. Thus, chemical oxidation after bioleaching made it possible improve the recovery of non-ferrous metals from sulfide concentrates only in the presence of a high concentration of sodium chloride.

Despite an increase in copper extraction, it was less than 50%. This may be explained by the resistance of tennantite to oxidative leaching. Copper extraction probably occurred mainly due to chalcopyrite leaching as this process depends on both chloride presence and temperature [3,16,17,33,47,48], while tennantite leaching may be inhibited by chloride ions and depends on temperature to a lesser extent [33,46]. This may be supported by the fact that the arsenic content in all chemical leaching residues was higher than that in the concentrate (3.70, 1.95, and 2.35% in the residue obtained after the leaching without NaCl, in the presence of 0.5 M NaCl, and 1 M NaCl, respectively) and by comparatively high tennantite content in the leaching residues. It should be noted that the oxidizing agent (ferric ions) was not consumed completely in all variants of the leaching test, while ferrous iron concentrations at the end of the experiments did not differ significantly, although the rate of ferrous ion formation in the beginning of the leaching was higher in the variants with NaCl leaching (Figure 4). This means that the rate of oxidation of copper minerals with ferric ions was not significant after 9 h of leaching.

The zinc concentration decrease may be explained by intensive jarosite precipitation especially in the presence of 0.5 M NaCl as jarosite formation may lead to zinc precipitation [57,58,59,60].

## 4. Discussion

The results obtained demonstrated that the proposed two-stage process made it possible to increase the bioleaching efficiency, as chemical oxidation can allow to perform the oxidation of refractory sulfide minerals under conditions that promote efficient oxidation of the minerals (high temperature, addition of sodium chloride), but can suppress the activity of microorganisms that carry out biooxidation [16]. Biooxidation in both temperature modes did not provide a high degree of copper extraction due to the resistance of copper minerals to biooxidation. While a temperature of 90 °C and high concentrations of sodium chloride, which provided an increase in copper extraction, are not suitable for microorganism activity [18,19,61].

It should be noted that the chalcopyrite leaching rate increase in the presence of chloride ions is a well-known fact [47,48]. As brines and marine water are often used in copper hydrometallurgy under conditions of water shortage [45,46,49] and allow the performance of chloride leaching of chalcopyrite ores and concentrates, they may also be used in the scheme proposed to supplement the leach solution. Although the presence of chloride ions in pregnant leaching solutions containing copper and zinc may impede further metal extraction, the use of appropriate conditions and reagents allows both copper and zinc extraction from the solutions containing chloride [62,63,64,65].

At the same time, biooxidation at the first stage of the process made it possible to extract some of the metals and to obtain solutions with a high concentration of a strong oxidizing agent, ferric ions. It has been shown in a number of studies that one-stage ferric leaching of various raw materials (slags, sulfide concentrates) using solutions containing ferric sulfate makes it possible to effectively extract non-ferrous metals into solution [66,67,68,69]. However, the problem is obtaining solutions of ferric ions in amounts sufficient for the oxidation of mineral raw materials. Therefore, “one-stage” ferric leaching of sulfide should either include a separate stage of biological or chemical regeneration of ferric ions or ferric-iron-containing reagents should be supplemented. The scheme proposed in the present work can make it possible to produce an oxidant solution from the products subjected to bioleaching and then perform oxidation at the second stage under conditions that allow leaching of metals to be carried out to the greatest extent. At the same time, the scheme proposed in the present work should not require application of additional oxidizing agents such as ferric sulfate/chloride. The same reason may explain the advantages of the approach proposed over the application of alternative leaching agents (ammoniacal lixiviants, ionic liquids, glycine, hydrogen peroxide, MnO_2_, nitrate) as it allows to avoid the necessity to produce, transport, store, and use additional reagents, as well as to apply additional equipment [44,45].

Despite this, in the present work, we were faced with the problem of a low tennantite bioleaching rate, which was also observed in our previous work on the leaching of similar concentrates containing chalcopyrite, tennantite, and sphalerite [1,18,32,48]. In our previous work [70], we proposed a two-stage leaching including stages of alkaline sulfide leaching (ASL), which allowed the transformation of tennantite refractory to bioleaching into copper sulfides, and continuous bioleaching. The proposed scheme made it possible to increase copper extraction by means of tennantite destruction. Thus, the possibility of the development of combined processes including stages of ASL, bioleaching, and chemical leaching at a high temperature should be studied to solve the problem of the leaching of complex copper–zinc concentrates.

## 5. Conclusions

In the present study, we attempted to modify the bioleaching process by introducing an additional stage to increase metal recovery and to develop technology that includes stages of biological and chemical leaching, which are performed under different conditions. The continuous bioleaching allowed extraction of 29.5% of copper and 78% of zinc and to obtain a pregnant solution, which may be used for further processing using oxidative leaching. The solid bioleach residue was then processed at a high temperature by means of leaching at 90 °C for 25 h in the presence of 1 M NaCl using the liquid phase of the bioleaching pulp as the leach solution. Cu and Zn extraction were increased up to 48 and 84%. Thus, two-stage leaching may allow the increase of bioleaching efficiency by means of leaching refractory minerals under optimum conditions (high temperature and chloride ion concentration). The scheme proposed in the present work can make it possible to produce an oxidant solution from the products subjected to bioleaching and then perform oxidation at the second stage under conditions that allow the leaching of metals to be carried out to the greatest extent. Therefore, the scheme proposed in the present work should not require application of additional oxidizing agents such as ferric sulfate/chloride.

## Figures and Tables

**Figure 1 microorganisms-10-01781-f001:**
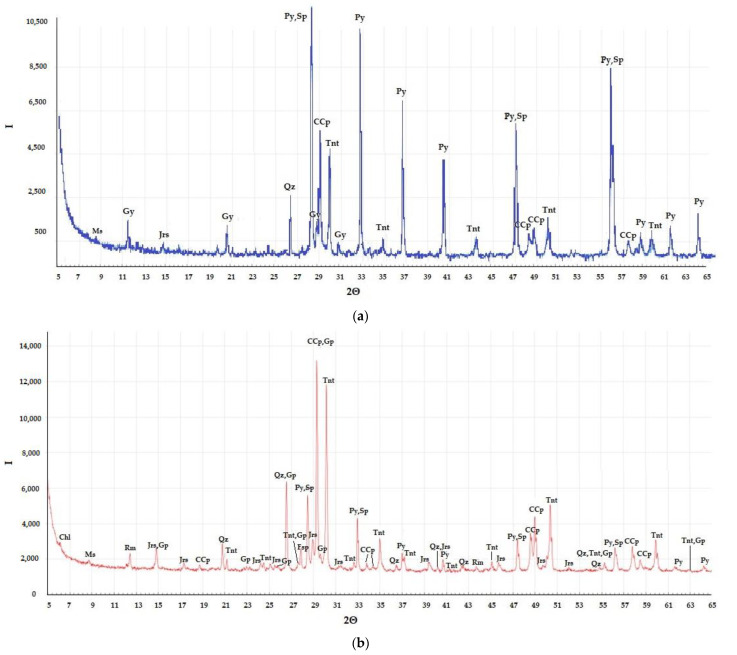
X-ray diffraction pattern of the samples: (**a**) concentrate; (**b**) bioleaching residue (1 mode); (**c**) bioleaching residue (2 mode); (**d**) ferric leaching residue (90 °C); (**e**) ferric leaching residue (90 °C, 0.5 M NaCl); (**f**) ferric leaching residue (90 °C, 1.0 M NaCl). Py: pyrite; Sp: sphalerite; Tnt: tennantite; Ccp: chalcopyrite; Gp: gypsum; Qz: quartz; Jrs: jarosite; Rm: ramsbeckite; Fsp: feldspar; Ms: mica; Chl: chlorite; Cer: cerussite.

**Figure 2 microorganisms-10-01781-f002:**
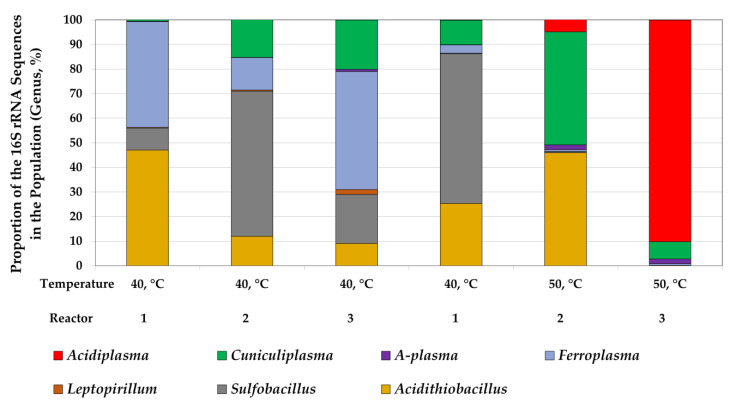
Analysis of microbial populations performing bioleaching under different conditions (proportion of the 16S rRNA gene fragment).

**Figure 3 microorganisms-10-01781-f003:**
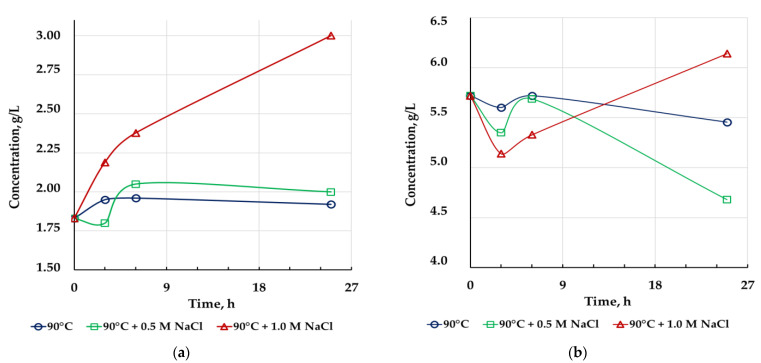
Changes in copper (**a**) and zinc (**b**) concentrations during chemical leaching of bioleaching residues under different conditions.

**Figure 4 microorganisms-10-01781-f004:**
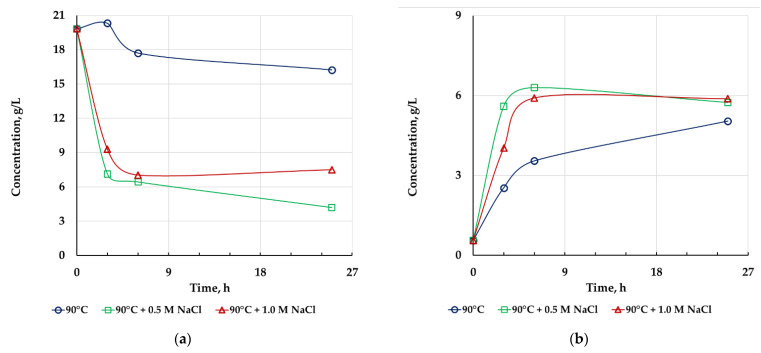
Changes in Fe^3+^ (**a**) and Fe^2+^ (**b**) ion concentrations during chemical leaching of bioleaching residues under different conditions.

**Table 1 microorganisms-10-01781-t001:** Main elemental contents in the concentrate sample.

Component	Content, %
SiO_2_	4.02
Al_2_O_3_	1.32
CaO	1.40
S_total_	38.6
Fe_total_	24.4
As	1.36
Cu	6.22
Zn	7.30
Pb	0.74

**Table 2 microorganisms-10-01781-t002:** Average values of liquid phase parameters during bioleaching experiments (steady state conditions).

Mode	Reactor	Residence Time, Day	T, °C	pH	Eh, mV	Fe^3+^, g/L	∑Fe, g/L	As, g/L	Cu, g/L	Zn, g/L
1	1	3	40	1.01 ± 0.06	749 ± 10	15.1 ± 0.3	15.2 ± 0.4	0.42 ± 0.02	0.96 ± 0.06	5.92 ± 0.38
	2	2	40	0.83 ± 0.06	776 ± 11	21.3 ± 0.3	21.3 ± 0.4	0.58 ± 0.03	1.29 ± 0.05	6.63 ± 0.27
	3	2	40	0.82 ± 0.05	843 ± 17	24.2 ± 1.1	24.2 ± 1.1	0.61 ± 0.05	1.62 ± 0.05	6.50 ± 0.24
1	1	3	40	0.92 ± 0.06	773 ± 9	15.8 ± 1.0	15.8 ± 1.0	0.44 ± 0.05	0.91 ± 0.08	5.72 ± 0.45
	2	2	50	0.69 ± 0.03	794 ± 5	20.8 ± 1.9	20.8 ± 1.9	0.50 ± 0.06	1.40 ± 0.09	5.82 ± 0.42
	3	2	50	0.67 ± 0.05	824 ± 13	23.4 ± 1.7	23.4 ± 1.7	0.57 ± 0.05	1.83 ± 0.10	5.72 ± 0.33

**Table 3 microorganisms-10-01781-t003:** Metal extraction into the liquid phase during bioleaching.

Mode	Reactor	Residence Time, Day	T, °C	Extraction, %	
				Cu	Zn
1	1	3	40	15.0 ± 0.9	81.1 ± 5.2
	2	2	40	20.8 ± 0.7	90.9 ± 3.8
	3	2	40	26.0 ± 0.8	89.0 ± 3.4
2	1	3	40	14.6 ± 1.4	78.4 ± 6.1
	2	2	50	22.5 ± 1.4	79.8 ± 6.1
	3	2	50	29.5 ± 1.6	78.3 ± 4.5

**Table 4 microorganisms-10-01781-t004:** Metal extraction into the liquid phase during chemical leaching of bioleaching residues.

Chemical Leaching Conditions	Extraction, %	
	Cu	Zn
90 °C	30.5	74.7
90 °C + 0.5 M NaCl	32.2	64.1
90 °C + 1.0 M NaCl	48.2	84.1

## Data Availability

Not applicable.

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
