# Peer review of "Two-Stage Oxidative Leaching of Low-Grade Copper–Zinc Sulfide Concentrate"

_microorganisms, 2022, doi:10.3390/microorganisms10091781_

Round 1

Reviewer 1 Report

Dear Authors

This manuscript called “Two-stage oxidative leaching of low-grade copper-zinc concentrate” is very well written and structured. The results analyzes are also clearly understood in general. The introduction should simply be improved a little more (complement), and the conclusions should be improved in general.

In the first paragraph of the manuscript, the environmental problems generated by the foundry plants are mentioned. However, the biggest environmental problem is the tailings generated in flotation processes, and none of this is mentioned. For example “In large-scale copper mining, for every ton of copper produced, approximately 150 tons of tailings are generated” “Recent estimates for global tailings volumes total over 44.5 billion m3 in nearly 2000 storage facilities, and projections total over 55.8 billion m3 by the year 2025” DOI: 10.1016/j.mineng.2021.106814

In the technologies for leaching processes, the incorporation of alternative waters to the processes, such as the use of seawater or wastewater, is not mentioned either.

Improve the conclusions, and mention them point by point by finding.

Regards

Author Response

This manuscript called “Two-stage oxidative leaching of low-grade copper-zinc concentrate” is very well written and structured. The results analyzes are also clearly understood in general. The introduction should simply be improved a little more (complement), and the conclusions should be improved in general.

Authors are grateful to the Reviewer for evaluation of the work. Please, find our answers below:

Questions and answers:

1. In the first paragraph of the manuscript, the environmental problems generated by the foundry plants are mentioned. However, the biggest environmental problem is the tailings generated in flotation processes, and none of this is mentioned. For example “In large-scale copper mining, for every ton of copper produced, approximately 150 tons of tailings are generated” “Recent estimates for global tailings volumes total over 44.5 billion m3 in nearly 2000 storage facilities, and projections total over 55.8 billion m3 by the year 2025” DOI: 10.1016/j.mineng.2021.106814

Indeed, tailings and slag formation during copper production is one of the main problems, therefore additional references and facts were added in in the manuscript. In the same time, our results are not directly focused on the issue of wastes treatment and our research is dedicated to treatment of sulfide concentrate, i.e., it does not solve problem of tailings formation during ore dressing. Therefore comprehensive  information on the issue of tailings production, waste storage, etc. was not included in the introduction.

2. In the technologies for leaching processes, the incorporation of alternative waters to the processes, such as the use of seawater or wastewater, is not mentioned either.

As in our study we have used leach solution with high chloride content, an additional references were included and additional information regarding chloride-containing waters application as well as treatment of PLS with high chloride content was added in the discussion.

3. Improve the conclusions, and mention them point by point by finding.

The conclusion has been improved.

Reviewer 2 Report

Two-stage oxidative leaching of low-grade copper-zinc concentrate is very interesting paper! Some improvement is required!

Page 1: Title:

Two-stage oxidative leaching of low-grade copper-zinc concentrate

Can you to write:  Two-stage oxidative leaching of low-grade copper-zinc sulfide concentrate

Page 4: One of the challenges for applying bioleaching to these concentrates is the fact that many copper sulfide minerals, such as chalcopyrite and tennantite, are relatively resistant to bioleaching. Is it possible to develop one selective leaching process (without Iron Extraction)

Page 4: Is two-stage hydrometallurgical scheme one continuous process?

Page 7: in 1 L laboratory reactors (how many laboratory reactors?)

Page 7: What ist the leaching agent for the chemical leaching proces

Page 8: In the samples in liquid phase, pH and redox potential (Eh) were determined using a pH-150MI pH meter (Izmeritelnaya tekhnika, Moscow, Russia), and ferrous and ferric iron were measured by trilonometric titration. At which temperature and time are these measurements performed?

Page 8: Please write an unit for Residence time in Table 2.

Page 9: plase to write an unit for Residence time in Table 3. Metal extraction into liquid phase during bioleaching

Page 9: Thus, temperature increase led to the increase in copper extraction, probably due to chalcopyrite leaching rate, while zinc extraction at higher temperature decreased. Why? Is precipitation of zinc present at higher temperature and zinc is removed from Solution?

General questions:

1.     is possible cementation of copper with zinc in solution?

2.     In Table 4. Metal extraction into liquid phase during chemical of bioleaching residues, the leaciching efficiency of copper is only 30 %. Why did you not use H2O2 in order to improve leaching efficiency of copper?

3.     Is possible to perform only one-stage oxidative leaching of low-grade copper-zinc concentrate at 90 °C instead Two-stage oxidative leaching of low-grade copper-zinc concentrate

Author Response

Two-stage oxidative leaching of low-grade copper-zinc concentrate is very interesting paper! Some improvement is required!

Authors are grateful to the Reviewer for evaluation of the work. Please, find our answers below:

Questions and answers:

  1. Page 1: Title: Two-stage oxidative leaching of low-grade copper-zinc concentrate

Can you to write:  Two-stage oxidative leaching of low-grade copper-zinc sulfide concentrate

The title was changed.

  1. Page 4: One of the challenges for applying bioleaching to these concentrates is the fact that many copper sulfide minerals, such as chalcopyrite and tennantite, are relatively resistant to bioleaching. Is it possible to develop one selective leaching process (without Iron Extraction).

Indeed, various alternative approaches for copper leaching (ammonia, glycine, ionic liquids) has been developed, which are discussed, for example, in the review of Barton and Hiskey, 2022. In the same time, leaching of copper concentrates in different ferric systems still considered as promising approach due to comparative cheapness and simplicity of ferric reagents as well as due to well-studied mechanisms, well-known processes of further copper extraction from PLS (solvent extraction, cementation) and possibility to extract other valuable metals simultaneously with copper. Simultaneous leching of copper and iron from the ores and concentrates pose a certain problem as iron is usually not valuable product in copper extraction process. Despite this, it is known that copper (and other metals) may be extracted from iron-containing solutions using solvent extraction. In our study, we have used bioleaching not only for copper extraction but also to obtain ferric ion containing solutions for further chemical leaching. Additional explanations have been added in the introduction and into the purpose of the work.

  1. Page 4: Is two-stage hydrometallurgical scheme one continuous process?

Additional explanations have been added in the introduction and into the purpose of the work.

The purpose of this work was to conduct laboratory tests on tank bioleaching of a copper-zinc concentrate containing chalcopyrite, tennantite, sphalerite, and pyrite, as well as to develop a two-stage hydrometallurgical scheme, including (1) bioleaching stage for copper and zinc extraction and obtaining solution containing leaching agent (ferric ions) and (2) high-temperature chemical oxidation stage to increase metal extraction degree. According to the proposed scheme, these two stages may be used as sequential stages of integrated continuous leaching process.

  1. Page 7: in 1 L laboratory reactors (how many laboratory reactors?)

Bioleaching tests were carried out in 3 reactors connected in series (page 7).

  1. Page 7: What ist the leaching agent for the chemical leaching process

Thus, as liquid phase of bioleaching pulp contained Fe3+ ions, which is the strong oxidant, and, in some cases, was supplemented with NaCl, ferric and chloride ions were leaching agents in chemical leaching process (Page 8).

  1. Page 8: In the samples in liquid phase, pH and redox potential (Eh) were determined using a pH-150MI pH meter (Izmeritelnaya tekhnika, Moscow, Russia), and ferrous and ferric iron were measured by trilonometric titration. At which temperature and time are these measurements performed?

To perform measurements, samples were analyzed after collection and cooling to ambient temperature. pH meter also possessed thermocompensation.

The pH and redox potential (Eh) were measured in liquid phase using a pH-150MI pH meter (Izmeritelnaya tekhnika, Moscow, Russia) with thermocompensation (Page 8).

  1. Page 8: Please write an unit for Residence time in Table 2.

The unit (day) was added.

  1. Page 9: plase to write an unit for Residence time in Table 3. Metal extraction into liquid phase during bioleaching

The units (day and %) were added.

  1. Page 9: Thus, temperature increase led to the increase in copper extraction, probably due to chalcopyrite leaching rate, while zinc extraction at higher temperature decreased. Why? Is precipitation of zinc present at higher temperature and zinc is removed from Solution?

Based on the results obtained, the reliable explanation of zinc recovery decrease during the bioleaching at higher temperature may not be obtained. In out previous work (doi.org/10.3390/min12050592), we have observed the same effect during the bioleaching of similar concentrate at different temperature. Probably, this effect may be explained by similar trends in the changes in microbial population composition (the decrease in Sulfobacillus and Acidithiobacillus fraction and the increase in the proportion of archaea) when increasing the temperature from 40 to 50°C. However, as information on the effect of the change in microbial population composition on sphalerite leaching is absent, this explanation may be used only as assumption.

This explanation was added in the manuscript.

Zinc removal from the solution due to jarosite precipitation cannot be the explanation as zinc concentration did not decrease in the reactors, while iron ions concentrations increased in both experiments in contrast to chemical leaching experiments, in which ferric and total iron concentration decreased in the presence of NaCl due to jarosite formation.

  1. is possible cementation of copper with zinc in solution?

We have considered possibility of copper and zinc solvent extraction from the solutions in the Discission and added several references regarding copper and zinc extraction from PLS with high chloride content.

  1. In Table 4. Metal extraction into liquid phase during chemical of bioleaching residues, the leaciching efficiency of copper is only 30 %. Why did you not use H2O2 in order to improve leaching efficiency of copper?

Indeed, additional oxidants application may improve and this assumption has been included in the discussion. In the same time, it should be noted that after chemical leaching PLS contained some amounts of the oxidant (ferric iron), which was not reduced completely. Thus, copper mineral leaching from leaching residues is questionable. Moreover, in the present study we made an attempt to perform concentrate leaching without additional leaching reagent addition and evaluate its efficiency, while modification of the scheme proposed may be performed in further research.

  1. Is possible to perform only one-stage oxidative leaching of low-grade copper-zinc concentrate at 90 °C instead Two-stage oxidative leaching of low-grade copper-zinc concentrate.

Indeed, one stage ferric leaching of similar concentrates has been studied. In the present work, we used bioleaching to avoid application of additional reagents (ferric sulfate/chloride) as bioleaching made it possible to obtain solution with high ferric ion concentration.

Additional explanations have been added to the purpose of the work.

Round 2

Reviewer 1 Report

Dear Authors

Hello, the suggested changes were made and the manuscript improved considerably. I recommend this manuscript for publication.

Regards